# EFFICIENT EVALUATION OF LARGE LANGUAGE MODELS VIA COLLABORATIVE FILTERING

## ABSTRACT

With the development of Large Language Models (LLMs), numerous benchmarks have been proposed to measure and compare the capabilities of different LLMs. However, evaluating LLMs is costly due to the large number of test instances and their slow inference speed. In this paper, we propose a collaborative filtering–inspired method that estimates model performance on a benchmark using only a small subset of test instances. Specifically, we treat "LLM–instance" interactions as "user-item" interactions and design a two-stage approach. Our method first selects a small set of representative instances for a given task and then predicts the overall task-level performance from the model's results on these selected instances. These two stages correspond to the cold-start problem and the rating prediction problem in recommendation systems, respectively. Experiments on multiple LLMs and benchmarks demonstrate that our method achieves performance estimation 3% error using 10% of test data, reducing evaluation cost by an order of magnitude while maintaining high accuracy.

## 1 INTRODUCTION

Large Language Models (LLMs) have garnered widespread attention, with numerous LLMs (Bai et al., 2023; Touvron et al., 2023a;b; Zeng et al., 2023) being released and rapidly iterated. Due to their powerful general capabilities, these LLMs are expected to perform a diverse and broad range of tasks (Zhou et al., 2023; Pham et al., 2024; Gao et al., 2023; Qin et al., 2023). To fairly assess and compare different LLMs' capabilities, many benchmarks for evaluating LLMs have emerged and developed continuously (Liu et al., 2024; Hendrycks et al., 2021; Li et al., 2024a; Zhong et al., 2024; Liang et al., 2023). A well-designed benchmark often requires a large number of test instances, because it is constructed to comprehensively and accu-

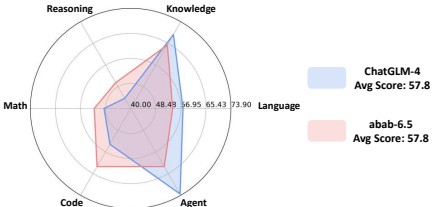

Figure 1: **OpenCompass Scores of ChatGLM-4 (Zhipu AI) and abab-6.5 (MiniMax).** Although they share the same average score, the LLMs exhibit substantial performance differences across individual tasks.

rately evaluate the diverse capabilities of LLMs. Such benchmarks typically cover a wide range of tasks (or scenarios), with each task corresponding to a sub-dataset. Moreover, even within a specific task, such as code generation (Lu et al., 2021), multiple programming languages may be involved, each with its own sub-dataset.

Given the large number of test instances and the relatively slow inference speed of LLMs, it is costly to let LLMs infer on all instances to obtain their performance. For example, running HELM benchmark can entail spending $10K+ or 4K+ GPU hours for evaluating a single model (Liang et al., 2023), and may even surpass those of pretraining (Biderman et al., 2023) when evaluating checkpoints. To address this, we aim to accurately evaluate the capabilities of an LLM on each individual task at a low cost.

Although several prior studies (Bommasani et al., 2021; Prabhu et al., 2024; Polo et al., 2024) attempt to estimate the overall performance of LLMs by performing inference on only a subset of test instances, these methods primarily focus on aggregate accuracy and overall rankings, while overlooking the more challenging problem of task-level performance estimation. In fact, compared with

overall metrics, task-level accuracy and ranking are substantially more difficult to estimate: although prior methods often achieve good performance on global indicators, their estimates for individual tasks are highly susceptible to sampling bias. As shown in Figure 1, models with similar overall scores may still exhibit large discrepancies on specific tasks, further illustrating the limitations of existing approaches. Our experimental results (see in section 5) also confirm this observation—current methods often perform poorly at the task level and can even underperform the simple baseline introduced in subsection 3.3, failing to reveal the true capability differences of models on different tasks. Consequently, existing efficient evaluation techniques remain insufficient for practical scenarios, where fine-grained, task-level capability assessment is typically required. To the best of our knowledge, this work is the first to systematically emphasize and address the estimation of task-specific performance and ranking, highlighting the necessity of developing low-cost, task-aware evaluation methods that can more accurately capture the capabilities of LLMs across diverse tasks.

To estimate the performance of LLMs on each task with low cost, we propose a two-stage method inspired by Collaborative Filtering (CF) in Recommendation Systems (RS). This method is tested on benchmarks containing various tasks and achieves more accurate estimates of model performance on each individual task at a low cost compared to previous methods (Prabhu et al., 2024; Polo et al., 2024). Figure 2 shows an example from a randomly selected task in MMLU benchmark along with the workflow of our method, where it can be seen that our method consistently outperforms the others. The experimental details are consistent with those described in section 5.

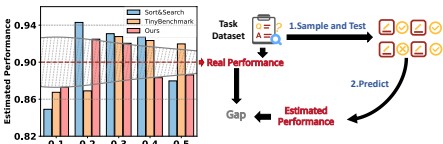

Figure 2: **Comparison between Methods (left) and Problem Setting (right).** Left: red line is real performance of target model, gray area is performance gap. Right: goal is to select subsets to minimize estimation error.

In our method, we treat LLMs as users and instances as items. The two-stage process focuses on determining the value of all items for a new user by utilizing both the interaction history of other users with items and the new user's interaction history with some items. In the first stage, we draw inspiration from the cold-start problem in recommendation systems. When user information is insufficient, a common strategy is to recommend popular items to observe user preference patterns. Similarly, we select a small set of representative instances based on importance scores, for example, about 10% of instances in each scenario, and obtain the target model's results on these instances as the basis for the subsequent stage. In the second stage, we view performance prediction as rating prediction problem in RS to predict the target LLM's behavior on unselected instances. Specifically, we predict performance using CF, based on the results of similar LLMs on remaing 90% unselected instances and the results of the target LLM on selected instances and synthesized instances by optimal transport (Peyré & Cuturi, 2019). Our contributions are as follows:

- We propose an efficient evaluation method based on the idea of collaborative filtering, which can efficiently give the performance of LLMs on different tasks.

- We analyze the similarities and differences between efficient evaluation and Recommendation Systems, which inspire us to apply methods of RS to address the efficient evaluation problem.

- Experiments on various benchmarks demonstrate the effectiveness of our method.

## 2  RELATED WORK (MORE IN APPENDIX A)

**Efficient Evaluation Methods of PFMs.** With the rise of Pre-trained Foundation Models (PFMs), many benchmarks have been proposed to quantify and compare model capabilities. However, growing model and dataset sizes have made evaluation increasingly costly. To address this, various efficient evaluation methods (Perlitz et al., 2024; Liang et al., 2023; Vivek et al., 2024; Polo et al., 2024; Prabhu et al., 2024; Zhou et al., 2025; Pacchiardi et al., 2024) have been proposed.

**Data Selection for LLM.** Some previous work (Schoch et al., 2023; Xie et al., 2023b;a) has attempted to select training data for LLM during the training phase to reduce the impact of low-quality training instances on model performance and improve training speed and efficiency. In this work, unlike data selection methods that select data during the training phase, we primarily focus on instance selection during the testing phase of large language models.

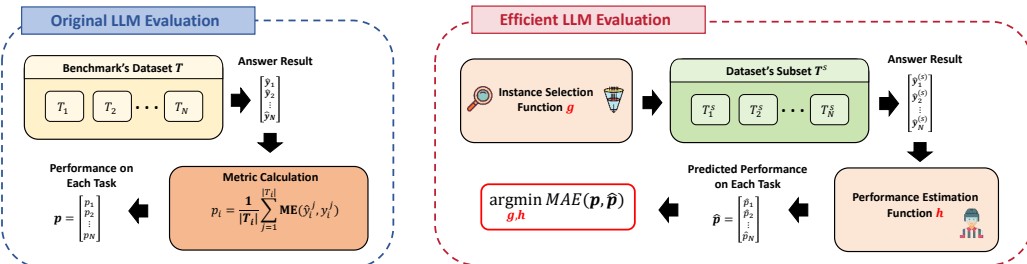

Figure 3: **The Paradigms of Original and Efficient LLM Evaluation.**. Left part implies the evaluation process of Original LLM Benchmark. Right part shows the process of efficient evaluation method, which consists of two main components: the Instance Selection Function $g$ and the Performance Estimation Function $h$. The goal of efficient evaluation methods is to design effective $g$ and $h$ to minimize the difference between real performance $\boldsymbol{p}$ and predicted performance $\hat{\boldsymbol{p}}$.

**Collaborative filtering** (CF), as a core technique in recommendation systems, infers preferences by leveraging user-item interaction data (Goldberg et al., 1992; Schafer et al., 2007; Su & Khoshgoftaar, 2009), with various methods including memory-based (Wang et al., 2006), matrix factorization (Luo et al., 2014), and neural network approaches (He et al., 2017; Wang et al., 2019). We find that the CF paradigm can also be effectively applied to efficient evaluation, with detailed similarities and differences discussed in Appendix B.

## 3 PRELIMINARY

Here we will introduce the process of original LLM evaluation and the paradigm of task-level efficient evaluation methods, as shown in Figure 3. We will also present the evaluation metrics for comparing these methods and insights from a simple baseline.

### 3.1 EVALUATION FOR LLMS

Assume an LLM Benchmark dataset $T = \{T_1, \ldots, T_N\}$ with $N$ task-specific datasets, where $T_i = (x_i^j, y_i^j)_{j=1}^{|T_i|}$ contains instance–label pairs. The LLM $f$ takes $x_i^j$ as input and outputs a prediction $\hat{y}_i^j$ after post-processing. The performance on task $T_i$ is computed via:

$$p_i = \frac{1}{|T_i|} \sum_{j=1}^{|T_i|} \text{ME}(\hat{y}_i^j, y_i^j), \tag{1}$$

with ME being a metric like Exact Match (Rajpurkar et al., 2016) or ROUGE (Lin, 2004). This yields a performance vector $\boldsymbol{p} = [p_1, \ldots, p_N]$. After evaluating all the LLMs in the model zoo, we can also obtain the ranking $\boldsymbol{r} = [r_1, \cdots, r_N]$ of the LLM $f$. We need $|T| = \sum_{i=1}^{N} |T_i|$ forward inferences to evaluate the LLM $f$ on LLM Benchmark $T$. With each inference taking an average of $t$ seconds, the total evaluation time is $|T| \times t$. Given the numerous test instances in LLM Benchmark and the relatively slow inference speed of LLMs, both $|T|$ and $t$ tend to be high, significantly increasing the time and computational resources required for evaluation. Our work aims to reduce the number of forward inferences $|T|$.

### 3.2 EFFICIENT EVALUATION METHOD FOR LLMS

Following the notation established in the previous section, LLMs can be ordered as $[f_1, f_2, f_3, \ldots]$ by release date. We assume evaluation results $D \in \mathbb{R}^{B \times |T|}$ are available for the earliest $B$ LLMs on all instances of benchmark $T$, where each entry is $\text{ME}(\cdot, \cdot)$, measuring answer quality. Higher values in $D$ indicate better performance. We refer to these $B$ LLMs as the initial model set $F_B$. We now focus on evaluating a new LLM on the Benchmark, mainly on its absolute performance and its ranking relative to the initial LLMs for each task. Previous methods require testing on all instances to compute ground-truth $p_i$ and $r_i$ (see Equation 1), whereas efficient evaluation seeks accurate estimates using only a subset of instances.

A well-designed efficient evaluation method consists of two core components: the **instance selection function** $g$ and the **performance prediction function** $h$. The instance sampling function $g$ leverages the evaluation results of initial LLMs $F_B$ on all instances for each task to select the important test instances. More specifically, for the $i$-th task $T_i$, a submatrix $D_i \in \mathbb{R}^{B \times |T_i|}$ containing only the evaluation results for the current samples can be extracted from $D$. The function $g$ then uses $D_i$ to select a subset $T_i^s$ from $T_i$:

$$T_i^s = g(D_i, T_i), \ |T_i^s| \ll |T_i|, \ T_i^s \subsetneq T_i. \tag{2}$$

Given the subset benchmark $T^s = \{T_1^s, \ldots, T_N^s\}$, the performance prediction function $h$ estimates the new model's performance $\hat{p}_i$ on task $i$ using $D_i$ and $T_i^s$:

$$\hat{p}_i = h(D_i, T_i^s). \tag{3}$$

Once $\hat{p}_i$ is obtained, it can be compared with the initial LLMs' performance $D_i$ to determine the new model's ranking $\hat{r}_i$ on each task.

Different efficient evaluation methods adopt different designs for the instance selection function $g$ and the performance prediction function $h$. To compare their effectiveness, we use the Mean Absolute Error (MAE) between predicted performances $\hat{\boldsymbol{p}} = [\hat{p}_1, \cdots, \hat{p}_N]$ and true performances $\boldsymbol{p}$:

$$\text{MAE}(\boldsymbol{p}, \hat{\boldsymbol{p}}) = \frac{1}{N} \sum_{i=1}^{N} |p_i - \hat{p}_i|. \tag{4}$$

Similarly, we compute the MAE between the predicted rankings $\hat{\boldsymbol{r}}$ and the ground-truth rankings $\boldsymbol{r}$. To ensure fairness, all methods are evaluated with the same number of selected instances $|T^s|$. A lower MAE indicates higher accuracy of the efficient evaluation method. Note that we also report additional metrics, such as Top-k Recall and NDCG, which are presented in Appendix G.

A well-designed efficient evaluation method should satisfy the following criteria: **1) High Efficiency.** It should generate accurate predictions quickly to meet the low-cost requirements in section 1. **2) Low Overhead.** The method should be easy to deploy, requiring minimal additional resources for application to new scenarios. **3) Commonness.** To estimate overall model performance with few samples, the method should reduce MAE (see Equation 4) by filtering out redundant instances and selecting informative ones. **4) Personalization.** A good benchmark (OpenCompass, 2023; HuggingFace, 2024) should reflect both absolute and relative performance (e.g., ranking $\boldsymbol{r}$), so the method should consider the differences of each model to give an accurate ranking between models. **5) Complementation.** Since some tasks share similar capabilities, the method should leverage information from other tasks.

### 3.3 INSIGHTS FROM A SIMPLE BASELINE

In this subsection, we introduce a simple yet effective efficient evaluation method as a baseline. Specifically, our baseline explores two types of embeddings: (i) semantic embeddings, extracted using off-the-shelf models such as Sentence-BERT (Reimers & Gurevych, 2019), and (ii) historical performance embeddings, which summarize past behaviors of existing models on benchmark tasks. As demonstrated in the subsequent toy experiment, the results show that naively relying on semantic similarity provides limited benefit and fails to yield performance improvements. In contrast, historical performance embeddings offer a much more informative signal.

In this baseline, we assume that the model produces consistent responses for similar samples. Specifically, for task $T_i$, we implement the instance selection function $g$ by applying K-means clustering on instance embeddings, and identifying the sample closest to the center of each cluster as the selected sample. This helps us identify groups of similar samples $\{C_{i1}, ..., C_{iK}\}$ along with their most representative instances $\{x_{i1}, ..., x_{iK}\}$. For the performance prediction function $h$, we assume that the model produces consistent responses for samples within the same cluster, and use the following weighted sum to get the estimated performance $\hat{p}_i$ for task $T_i$:

$$\hat{p}_i = \frac{\sum_{j=1}^{K} |C_{ij}| f(x_{ij})}{\sum_{j=1}^{K} |C_{ij}|} \tag{5}$$

Here, $|C_{ij}|$ is the number of instances in the $j$-th cluster of task $i$, and $x_{ij}$ is the instance closest to the cluster center.

We implement two common instance embedding methods for clustering: semantic embeddings from Sentence-BERT (Reimers & Gurevych, 2019), and the historical evaluation results of initial LLMs (i.e., $D_i$ from subsection 3.2). A toy experiment on a subset of the OpenCompass

Table 1: **Embedding Performance.**

| Method | Perf. MAE ↓ | Rank MAE ↓ |
|---|---|---|
| Historical | 0.035 | 2.9 |
| Semantic | 0.210 | 6.7 |

Benchmark (OpenCompass, 2023) compares the two methods, with results shown in Table 1. The method using historical evaluation results outperforms that using semantic embeddings. We hypothesize that semantic embeddings are less aligned with the evaluation space, while historical results better capture instance difficulty, making it easier to predict the performance on unselected instances. Experimental details can be viewed in Appendix C.

Beyond the weaker performance, using semantic embeddings presents several challenges: **High computational cost**: At least one embedding model infers on all the instances causing high consumption. **Large storage requirements**: Semantic embeddings and embedding model require more space to store than historical evaluation data. **Privacy concerns**: Some benchmarks only release partial datasets, and asking model developers to submit their models raises fairness and security issues. Based on the above considerations, our method relies solely on historical evaluation results rather than semantic embeddings.

## 4 METHOD

This section introduces a two-stage method based on Collaborative Filtering (CF), demonstrating its advantages over prior approaches. By treating LLMs as users and test instances as items, we construct features from their interactions to compute similarities—both between LLMs and between instances. In the first stage, we leverage these similarities to select discriminative instances. In the second stage, we use the selected instances to predict target model's performance on each task. Figure 4 illustrates the overall workflow of our method, and the pseudo-code is provided in Appendix E.

### 4.1 STAGE 1: SELECT TEST INSTANCES VIA CF

As described in subsection 3.2, the instance selection function $g$ aims to select important instances for each task. We first define the importance score, and then design an iterative sampling process for each task to select a small but discriminative subset of test instances, as shown in Figure 4.

#### 4.1.1 DEFINITION OF IMPORTANT SCORE

We first define the importance score of test instance $x$ for a given model set $F = \{f_1, \cdots, f_M\}$. Inspired by educational testing (der Linden & Glas, 2000), instances that are either too easy or too difficult offer little value, as most test takers tend to perform similarly. To effectively differentiate model capabilities, we prioritize instances where model performance varies significantly. Intuitively, the most valuable instances are those where some LLMs perform well while others perform poorly—leading to high variance in performance across the model set $F$. This idea is captured by the following definition of the importance score $v(x|F)$:

$$v(x|F) = \frac{1}{M-1} \sum_{m=1}^{M} \left( \text{ME}(\hat{y}^{(f_m)}, y) - \overline{\text{ME}}(\hat{y}, y) \right)^2.$$

(6)

where $\hat{y}^{(f_m)}$, $y$ and $\overline{\text{ME}}$ indicate the $m$-th LLM's answer, ground truth answer and average performance for instance $x$, respectively. The score represents the variability in the LLM's responses for a single instance. A higher value of this score indicates greater discriminative power of the instance.

#### 4.1.2 PROCESS FOR INSTANCE SAMPLING

To select important instances, two criteria should be met: (1) They should evaluate all LLMs, i.e., have high importance scores $v(x|F_B \cup f_t)$; (2) They should distinguish the target LLM $f_t$ from its similar models $F_S$, reflected by high scores $v(x|F_S \cup f_t)$.

Figure 4: **Steps in the Instance Selection Process and Performance Prediction Process.** In the Instance Selection Process, we select instances that can easily distinguish models through an iterative process. In the Performance Prediction Process, we predict model performance based on optimal transport and collaborative filtering.

To meet the above requirements, we design an iterative process. As Figure 4 left shows, given target LLM $f_t$, the $i$-th task $T_i$, initial models $F_B = \{f_1, \cdots, f_B\}$ and evaluation results of initial models $D_i \in \mathbb{R}^{B \times |T_i|}$, the process can select important instances with three steps.

**In the first step**, we construct a Probe Set $P$ by selecting instances from $T_i$ with high importance scores. For the $j$-th instance, its importance score $v(x_i^j | F_B)$ can be computed with the help of $D_i$ following Equation 6. This identifies instances that best differentiate the initial model performance.

**In the second step**, we let the target model $f_t$ infer on the Probe Set $P$, obtaining a result vector $d_t^P \in \mathbb{R}^{|P|}$. We then extract $D_{i,P} \in \mathbb{R}^{B \times |P|}$, the evaluation results of initial LLMs on $P$. By computing cosine similarities between $d_t^P$ and each row of $D_{i,P}$, we select the top-$n$ most similar LLMs to form $F_S$, with index set $S = \{k | f_k \in F_S\}$. The importance score $v(x_i^j | F_S)$ of instance $x_i^j$ on $F_S$ can be calculated following Equation 6. This helps identify instances that best distinguish the target model from its most similar peers.

**In the third step**, we combine $v(x_i^j | F_B)$ and $v(x_i^j | F_S)$ to compute the final importance score like Equation 7 thus meeting two conditions talked above.

$$v(x_i^j) = \alpha \cdot v(x_i^j | F_B) + (1 - \alpha) \cdot v(x_i^j | F_S). \tag{7}$$

We then select the top-$q$ instances by $v(x_i^j)$ to expand the Probe Set $P$, and repeat steps 2 and 3 until $|P|$ reaches the target size $|T_i^s|$, thereby obtaining the selected instance subset $T_i^s$ for task $T_i$. In this formula, the first term is fixed across target LLMs, while the second varies with different $F_S$, enabling personalized instance selection.

### 4.1.3 RELATIONSHIP WITH RECOMMENDATION SYSTEMS

The cold start problem in recommendation systems arises from insufficient interaction data for new users and items. A common solution is to initially recommend popular items to collect data (Luo et al., 2025; Jeon et al., 2024; Chaimalas et al., 2023), then apply collaborative filtering to personalize recommendations.

Similarly, we treat LLMs as users and instances as items. Our evaluation matrix $D_B$, reflecting LLM performance on instances, parallels the user-item interaction matrix but measures response quality rather than preferences. To recommend discriminative instances to a new LLM, we first construct a Probe Set of highly discriminative instances based on historical models' performance, analogous to recommending popular items. Based on the new LLM's results on the Probe Set, we identify similar models and select additional discriminative instances from them for further evaluation, following a user-based collaborative filtering approach.

### 4.2 STAGE 2: PREDICT LLM'S PERFORMANCE

After getting the subset benchmark $T^s = \{T_1^s, \cdots, T_N^s\}$, we need to design the performance prediction function $h$ to predict the performance $p_i$ and ranking $r_i$ of the new LLM.

### 4.2.1 Purpose of Performance Prediction

For performance prediction, a direct approach is to use the new LLM's accuracy $p_i^s$ on the sampled subset $T_i^s$. However, since selected instances exclude those too easy or hard, $p_i^s$ may not reflect the true performance $p_i$. To correct this, we revisit the formulation of $p_i$ in Equation 8.

$$p_i = \frac{a_i^s + a_i^{ns}}{|T_i^s| + |T_i^{ns}|}. \tag{8}$$

Here, $a_i^s$ and $a_i^{ns}$ are the sums of correct predictions on selected ($T_i^s$) and unselected ($T_i^{ns}$) instances. We know $a_i^s$, $|T_i^s|$, and $|T_i^{ns}|$, so predicting $a_i^{ns}$ allows us to estimate $p_i$ accurately.

### 4.2.2 Process For Performance Prediction

In collaborative filtering methods, a certain amount of historical interactions between new users and items is required to effectively estimate a new user's preferences for other items. However, in the efficient evaluation setting, where $|T_i^s| \ll |T_i^{ns}|$, collaborative filtering cannot be directly applied. To address this, we employ the Optimal Transport (OT) method to perform oversampling based on the selected instances from similar tasks, without relying on semantic information (see subsection 3.3). After oversampling, we subsequently apply collaborative filtering to obtain performance estimates of target model on unselected instances, thereby deriving estimated performance for each task.

**For the first step** in Figure 4 right, we first perform oversampling with OT to obtain oversampled instances, in order to expand the selected instances for subsequent collaborative filtering. Specifically, for the $i$-th task $T_i$, let $D_i \in \mathbb{R}^{B \times |T_i|}$ denote the evaluation results of initial LLMs $F_B$, and $d_t \in \mathbb{R}^{1 \times |T_i^s|}$ denote the performance of target model $f_t$ on $T_i^s$. We can get the mean performance vector $\mathbf{v}_i = \frac{\sum_{j=1}^{|T_i|} D_i^{:j}}{|T_i|}$ of $F_B$ for each task and calculate cosine similarity matrix $C_s \in \mathbb{R}^{N \times N}$. With the help of $C_s$ and threshold value $\tau_0$, similar tasks can be found and we can combine the selected data from similar tasks to get $T_i^{st}$ and $D_i^{(st)} \in \mathbb{R}^{B \times |T_i^{st}|}$. We can also extract a submatrix $D_i^{(ns)} \in \mathbb{R}^{B \times |T_i^{ns}|}$ which represents unselected samples from $D_i$. We then solve the Optimal Transport (OT) problem:

$$\underset{P \geq 0}{\text{argmin}} \quad \langle C_c, P \rangle \quad \text{s.t.} \quad \sum_j P_{jk} = \frac{1}{|T_i^{ns}|}, \quad \sum_k P_{jk} = \frac{1}{|T_i^{st}|} \tag{9}$$

where $C_c$ is a cost matrix defined by Euclidean distances between columns of $D_i^{(st)}$ and $D_i^{(ns)}$. With $P$, we can get oversampled data $T_i^o$ and corresponding matrix $D_i^{(o)} = D_i^{(st)} P$. By combining real target model responses on $T_i^s$ and oversampled data $T_i^o$, we effectively expand the interaction space of $f_t$, thereby enhancing the performance of subsequent collaborative filtering

Now we proceed to **the second step** in Figure 4 right, where, for a specifical task $T_i$, we apply collaborative filtering to estimate the new model's performance on the unselected instances, thereby obtaining the estimated performance for each task. For simplicity, we treat oversampled data as selected instances: $T_i := T_i^s \cup T_i^{ns} \cup T_i^o$ and $T_i^s := T_i^s \cup T_i^o$. Let $D_i^{(s)} \in \mathbb{R}^{B \times |T_i^s|}$ and $D_i^{(ns)} \in \mathbb{R}^{B \times |T_i^{ns}|}$ denote the feature matrices of selected and unselected instances.

To estimate the target model $f_t$'s performance on the unselected instances $T_i^{ns}$, we divide them into two parts based on instance importance (see in subsubsection 4.1.1) and similarity. This enables us to obtain a fine-grained estimation $\hat{c}_i^{ns}$ for the target performance $a_i^{ns}$ in Equation 8 on task $Ti$.

We use the importance scores to identify **the first part** and obtain the estimated performance of $f_t$ on it directly from the historical models $F_B$. Specifically, we assign importance scores in Equation 7 to $D_i^{(ns)}$ based on the average important score $\overline{v}$ from $D_i^{(s)}$, and use the average performance of $F_B$ as prediction $\hat{c}_{i0}^{ns}$ for instances below the threshold $\frac{\overline{v}}{\tau_1}$:

$$\hat{c}_{i0}^{ns} = \frac{1}{B} \sum_{k=1}^{B} \sum_{j \in S} D_i^{(ns)kj}, \quad T_i^{ns} := T_i^{ns} - S, \quad D_i^{(ns)} := D_i^{(ns)} - S \tag{10}$$

where $S = \{k \mid v_i^{(ns)k} < \frac{\overline{v}}{\tau_1}\}$. As noted in subsubsection 4.1.1, these instances are excluded as they lack discriminative value.

**For the second part**, we compute the cosine similarity matrix $C_i \in \mathbb{R}^{|T_i^{ns}| \times |T_i^s|}$ between $D_i^{(ns)}$ and $D_i^{(s)}$. For each unselected instance, we identify the top 3 most similar selected ones and calculate their average similarity $\bar{c}_i^j$. Using a threshold $\tau_3$, we split the unselected instances into two sets:

- For instances with $\bar{c}i^j \geq \tau_3$, we estimate performance using $f_t$'s results on the top-3 similar selected instances, yielding $\hat{c}_{i1}^{ns}$.
- For the rest, we use average results of LLMs similar to $f_t$, yielding $\hat{c}_{i2}^{ns}$.

Finally, we approximate $a_i^{ns}$ by $\hat{c}_i^{ns} = \hat{c}_{i0}^{ns} + \hat{c}_{i1}^{ns} + \hat{c}_{i2}^{ns}$ in Equation 8, obtaining the estimated performance $\hat{p}_i$, which outperforms $p_i^s$ in experiments. To predict LLM ranking, we compare $\hat{p}_i$ of the target LLM $f_t$ with the performances of initial LLMs to get the predicted ranking $\hat{r}_i$.

### 4.2.3 RELATIONSHIP WITH RECOMMENDATION SYSTEMS

The cold start problem in recommendation systems arises from insufficient interaction data for new users and items. A common solution is to transfer users' historical interaction data from other domains via cross-domain learning (Li et al., 2024b; Zhao et al., 2020; Guan et al., 2023), enabling personalized recommendations before enough local data is collected.

Similarly, our approach uses Optimal Transport (OT) to adapt new models' performance across tasks, effectively leveraging cross-domain information. We also incorporate both model-level and sample-level similarity, analogous to collaborative filtering based on user–item similarity.

### 4.3 COMPARISON WITH PREVIOUS METHODS

As shown in Table 2, our method meets the criteria described in subsection 3.2 which was not achievable with previous approaches. Specifically, our method only requires instances that are discriminative for models, which makes it **efficient** and **personalized**. For instances that are too simple or too difficult, our method uses the average results of the initial models for estimation. This ensures that our approach meets the **commonness** criterion. In addition, our method uses OT to leverage information from other tasks for prediction, meeting the **complementation** criterion.

Table 2: **Comparison of Evaluation Methods.**

| Aspect | Ours | Clust. | S&S | Tiny |
|---|---|---|---|---|
| **Efficiency** | ✓ | ✓ | ✓ | ✓ |
| **Low Overhead** | ✓ | ✓ | ✓ | ✗ |
| **Commonness** | ✓ | ✓ | ✓ | ✓ |
| **Personalization** | ✓ | ✗ | ✓ | ✗ |
| **Complementation** | ✓ | ✗ | ✗ | ✓ |

## 5 EXPERIMENTS

**Setups.** Our experiment focuses on efficiently evaluating new LLMs on Benchmarks using the results of some initial LLMs. This aligns with real-world scenarios. Based on release dates, we choose early LLMs for each Benchmark as initial models. As discussed in section 1, overall performance cannot capture task-level performance, so we assess whether the method can accurately predict per-task results. To compare the adaptability of the efficient evaluation method to different sample sizes, we set 5 ratios ([0.1, 0.2, 0.3, 0.4, 0.5]) to select corresponding subsets from each task's dataset.

**Benchmark.** We select three widely used LLM benchmarks. (1) HuggingFace Open LLM Leaderboard evaluates open-source LLMs on tasks such as understanding, generation, and reasoning. Following TinyBenchmark (Polo et al., 2024), we use results from 395 models, split 3:1 by release date. (2) MMLU, a popular 57-task multiple-choice QA benchmark. We use the same 395 models and splits, reflecting its prevalence in evaluations. (3) OpenCompass OpenCompass (2023) covers diverse tasks; we use 32 models across 187 scenarios, also split 3:1 by release date.

**Baselines.** We compare against four baselines: Random Sampling, Baseline with Clustering, TinyBenchmark, and Sort&Search. Baseline with Clustering is described in subsection 3.3 and omitted here. Random Sampling selects instances randomly from each task's dataset to form subsets. LLM performance on these subsets estimates its performance on the full dataset, and LLMs are ranked accordingly. For TinyBenchmark (Polo et al., 2024) and Sort&Search (Prabhu et al., 2024), the target LLM's estimated performance is compared with actual performances of other LLMs to produce rankings. For stochastic methods, we repeat the experiment 5 times and report the mean.

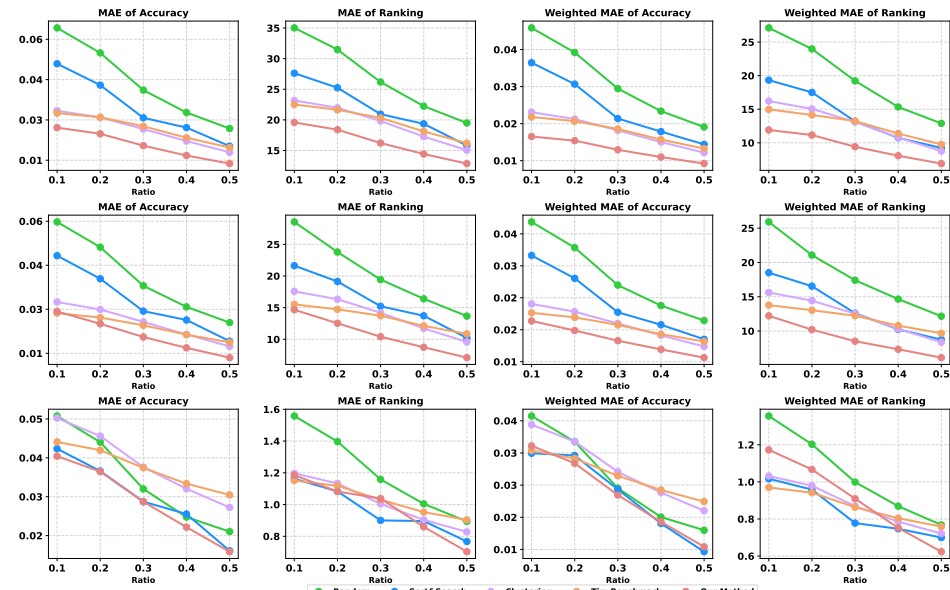

Figure 5: Mean Absolute Error (MAE) and weighted MAE between estimated and actual LLM performance by different methods. **Three rows show results on MMLU, LB, and OpenCompass.**

**Task-level Evaluations.** For each task, we calculate the Mean Absolute Error (MAE) between the predicted performance and the actual performance for each model. Then, we take the mean of MAE for all tasks as the final performance of the Efficient Evaluation Method. For example, for the MMLU Benchmark, we should calculate the average MAE for 395 test models on 57 tasks. To emphasize strong models, we also compute a weighted MAE using $\frac{1}{r_i}$ based on true rankings $r_i$. Metrics like Top-k Recall and NDCG are also reported in Appendix G.

**Results Analysis.** Figure 5 shows the Mean Absolute Error (MAE) and Weighted Mean Absolute Error (Weighted MAE) between the performance metrics (performance scores and rankings) of large language models (LLMs) estimated by different efficient evaluation methods and their actual metrics. Smaller values indicate more accurate estimations. As shown in Figure 5, our method achieves the most accurate estimates on Open LLM Leaderboard and MMLU. On the OpenCompass benchmark, no single method significantly outperforms the others. This is due to the relatively small number of models, which limits the availability of effective historical information and results in relatively small variations in the evaluation metrics. However, we still observe that our proposed method outperforms the others in terms of accuracy. In summary, our method performs comparably to other approaches when the number of historical models is limited, and outperforms them when more historical models are available. This makes it a promising method.

**Running Time Analysis.** Runtime is an important metric for evaluating efficiency. Assuming LLM inference time is constant across instances, we compare methods on evaluating one model on MMLU with a 0.1 sampling ratio. Table 3 reports Deployment Time, Selection Time, Prediction Time and Total Time (sum of Selection Time and Pre-

Table 3: **Comparison of Time Between Methods.**

| Method | DT (s) | ST (s) | PT (s) | TT (s) |
|---|---|---|---|---|
| **Random** | 0 | 0.0015 | 0.0008 | 0.0023 |
| **Sort&Search** | 0 | 6.05 | 0.0167 | 6.07 |
| **Clustering** | 0 | 26.8 | 0.232 | 27.0 |
| **Tiny Benchmark** | 309 | 26.8 | 3.36 | 30.2 |
| **Ours** | 0 | 0.713 | 24.5 | 25.2 |

diction Time). Our method achieves lower total time than TinyBenchmark and Clustering with better performance. Although slower than Random and Sort&Search, the modest time increase is justified by significantly better performance.

**Ablation Study.** We conduct an ablation study on the components and parameters of the method described in section 4. The specific roles and details of the parameters can be found in Appendix H. The experimental results show that optimal transport is an indispensable module in our method. The similarity-based collaborative filtering approach used in this paper achieves a good balance between accuracy and efficiency. Our method demonstrates strong robustness to different parameter settings.

In addition, the experiments reveal that the performance reported in section 5 is not yet optimal, suggesting that our method still has further potential for improvement.

We conduct an ablation study on the components and parameters of the method described in section 4. The specific roles and details of the parameters can be found in Appendix H, where we also provide additional analysis explaining why matrix factorization–based approaches are not adopted in our framework. The experimental results show that optimal transport is an indispensable module in our method. The similarity-based collaborative filtering approach used in this paper achieves a good balance between accuracy and efficiency. Our method demonstrates strong robustness to different parameter settings. In addition, the experiments reveal that the performance reported in section 5 is not yet optimal, suggesting that our method still has further potential for improvement.

## 6 CONCLUSION

In this work, we focus on designing an efficient evaluation method to evaluate the target LLM's task-level capacities at a low cost. We re-examine the issue from the perspectives of collaborative filtering in recommendation systems and propose a two-stage method, which includes instance selection stage and performance estimation stage. The experimental results across multiple LLMs and datasets demonstrate the effectiveness of our method.

## ETHICS STATEMENT

This work does not raise any direct ethical concerns. All experiments are conducted on publicly available datasets and do not involve human subjects, private information, or sensitive content. The proposed methods are intended for advancing research on parameter-efficient fine-tuning of large language models. Potential societal impacts are consistent with those of general LLM research, including both positive applications (e.g., lowering computational cost and energy consumption) and risks of misuse (e.g., generating harmful or misleading text). We encourage responsible use of our methods and adherence to ethical guidelines in AI research and deployment.

## REPRODUCIBILITY STATEMENT

We have taken several measures to ensure the reproducibility of our work. All datasets used in this paper are publicly available and properly cited. The experimental settings, including model architectures, hyperparameters, training steps, and evaluation metrics, are described in detail in the main text and Appendix. For key results, we report the average performance over multiple random seeds. In addition, we provide pseudo-code and algorithmic descriptions in the appendix for clarity. Source code and instructions for reproducing our experiments will be made available upon publication.

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

In the Appendix, we introduce more details about the Experiments.

## A  DETAILS OF RELATED WORK

**Efficient Evaluation Methods of PFMs.** With the rise of Pre-trained Foundation Models (PFMs), many benchmarks have been proposed to quantify and compare model capabilities. However, the increasing model size and dataset scale have made full evaluation prohibitively expensive. To mitigate this cost, a variety of efficient evaluation methods have been proposed, each with different sampling and performance extrapolation strategies. (Perlitz et al., 2024) reduces evaluation cost by sampling fewer examples in HELM (Liang et al., 2023), where it uses a coarse-to-fine tournament to estimate model rankings rather than absolute scores. (Vivek et al., 2024) accelerates classification evaluation by clustering samples according to model confidence, selecting representative samples from each cluster, and estimating overall performance through a weighted aggregation based on cluster sizes and the predictions of the selected samples. TinyBenchmark Polo et al. (2024) uses psychometric modeling (IRT) to identify a small set of highly informative instances, and extrapolates performance through learned instance difficulty and model ability parameters. Lifelong Benchmark Prabhu et al. (2024) (Sort&Search), for vision models, first ranks instance difficulty using prior predictions, then performs uniform sampling across the difficulty spectrum, and finally applies a simple extrapolatione. Zhou et al. (2025) introduces UCBE, a multi-armed bandit method that adaptively allocates more budget to promising models; its strategy corresponds to adaptive sampling, while its prediction relies on upper-confidence-bound estimation—but it primarily identifies the best model, making it closer to model selection than efficient evaluation. Pacchiardi et al. (2024) jointly uses historical model outputs, model embeddings, and text embeddings for sampling and prediction through a regression-style model, but this introduces high storage and computation cost. Since our experiments (section 5) show that historical scores alone are sufficient, we omit text embeddings.

Compared with previous efficient evaluation methods—which typically rely on clustering, difficulty sorting, or additional embedding models and therefore often overlook model-specific behavioral differences as well as complementary information across tasks—our approach introduces fundamental improvements in both the sampling and prediction stages. Existing methods usually treat each task in isolation and assume static difficulty patterns, making them unable to capture fine-grained performance differences between models or exploit structural information shared across tasks. Our method effectively overcomes these limitations. In the sampling stage, we leverage cross-model performance variability to identify highly discriminative instances and further personalize the selected subset by incorporating the similarity between the target model and historical models—an ability that is largely absent in clustering-, difficulty-, or IRT-based strategies. In the prediction stage, unlike methods that extrapolate solely within a single task, we employ Optimal Transport to incorporate information from related tasks and apply collaborative filtering to infer unobserved instance scores using both model–model and instance–instance similarity. This enables our method to achieve more accurate task-level performance estimation with low overhead, significantly outperforming approaches that rely only on embeddings, static difficulty assumptions, or task-isolated extrapolation.

**Data Selection for LLM.** Some previous work has attempted to select training data for LLM during the training phase to reduce the impact of low-quality training instances on model performance and improve training speed and efficiency. (Schoch et al., 2023) proposes TS-DSHAPLEY to utilize Shapley Values to filter out harmful training data, thereby improving the performance after model fine-tuning. (Xie et al., 2023b) designs Data Selection with Importance Resampling (DSIR) to select a tailored subset from the pretraining dataset for a specific target instance distribution, aiming to maximize the performance of the pre-trained model while adhering to a fixed compute budget. DSIR estimates importance weights in a reduced feature space for tractability and selects data with importance resampling according to these weights. (Xie et al., 2023a) leverages distributionally robust optimization (DRO) to tune the domain weights without knowledge of downstream tasks. These domain weights decide the mixture proportions of pretraining data domains. In this work, we primarily focus on instance selection during the testing phase of large language models.

**Collaborative Filtering** (CF) is a foundational technique in recommender systems, leveraging user-item interaction histories to infer preferences Goldberg et al. (1992); Schafer et al. (2007); Su & Khoshgoftaar (2009). Traditional methods include memory-based approaches (e.g., user-based Wang et al. (2006) and item-based CF) and model-based approaches such as Matrix Factorization (MF) Luo et al. (2014), which learn latent representations of users and items. Neural Collaborative Filtering (NCF) He et al. (2017) extends MF with deep networks to capture non-linear interactions. More recently, Graph Neural Networks (GNNs) have been applied to CF Wang et al. (2019), utilizing the user-item bipartite graph to model higher-order relations. In summary, Collaborative Filtering (CF) is a mature and effective method in recommendation systems. We find that it can be applied to efficient evaluation, and Appendix B illustrates the similarities and differences between two scenarios.

# B COMPARISON OF COLLABORATIVE FILTERING IN RECOMMENDATION SYSTEMS AND EFFICIENT EVALUATION

## B.1 SIMILARITIES

In collaborative filtering, the underlying assumption is typically that similar users exhibit similar behaviors on the same items. In other words, if two users show similar interests or preferences for certain items, they are likely to have similar preferences for other items as well. This assumption forms the theoretical foundation of collaborative filtering methods, enabling them to infer the unknown preferences of users based on the behaviors of known users, thus facilitating effective personalized recommendations.

To validate the applicability of this assumption in efficient evaluation scenarios, we conducted a series of experimental analyses on each sub-task of the MMLU benchmark. Specifically, for each sub-task $T_i$ and evaluation results of all LLMs $D_i$, we can compute the correlation coefficient $\rho_i^{jk}$ between rows $j$ and $k$ in $D_i$, which reflects the similarity between models $j$ and $k$ on task $T_i$. Similarly, we can concatenate the results of all sub-tasks except for $T_i$, and obtain the similarity $\hat{\rho}_i^{jk}$ between models $j$ and $k$ across all other tasks excluding $T_i$. Figure 6 below shows the results of one representative scenario.

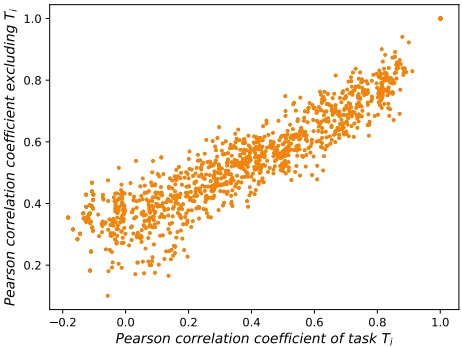

Figure 6: **The Scatter Plot of Pearson Correlation Coefficient.** Each point represents a pair of models. The x-axis denotes the Pearson correlation coefficient between the model pair on task $i$, while the y-axis denotes the Pearson correlation coefficient on tasks other than $i$. We observe a clear linear relationship between the x and y axes, indicating that similar models tend to produce similar responses.

From Figure 6, we can observe a clear linear relationship between the correlation coefficients, indicating that the basic assumptions of collaborative filtering in recommendation systems are also applicable in efficient evaluation.

## B.2 DIFFERENCES

In recommendation systems, the number of historical users is significantly higher than that in efficient evaluation tasks. However, we still believe that methods used in recommendation systems, such as collaborative filtering, remain applicable in the context of efficient evaluation. The reasons for this are as follows:

In recommendation systems, user-item interactions are typically sparse. This means that each user only interacts with a small subset of the available items. To make accurate recommendations, a large number of users is required, ensuring that the system can learn meaningful patterns and signals from the data effectively. The method employed must be capable of leveraging these interactions to understand users' preferences and predict future behavior.

On the other hand, in efficient evaluation tasks, interactions between historical models and evaluation instances are not sparse. This is because a historical model is usually evaluated on all the available evaluation instances, rather than just a small subset. Consequently, the number of models required in efficient evaluation is significantly lower compared to the number of users needed in recommendation systems. This difference arises because, in efficient evaluation, all instances can be assessed directly, whereas in recommendation systems, the model must generalize from limited interactions and data.

Furthermore, there is another key distinction between the two settings: In recommendation systems, the historical behavior of the target user is determined entirely by the user, and the system can only passively observe whatever interactions the user provides. In contrast, in efficient evaluation, the target model has not yet been evaluated, and it is up to us to actively decide which evaluation instances should be selected. Our method explicitly leverages this difference by actively choosing the most informative instances for evaluation, thereby substantially reducing the overall evaluation cost.

Therefore, while the scale and setup differ, the core assumptions of methods like collaborative filtering remain relevant in both contexts, and the ability to perform active selection in efficient evaluation makes these methods even more effective in our setting.

## C   EXPERIMENTS FOR CLUSTERING BASELINE

### C.1   EXPERIMENT SETTINGS

We sample 187 tasks Zellers et al. (2019); Clark et al. (2018); Bisk et al. (2020); Mihaylov et al. (2018); Lai et al. (2017); Xu et al. (2021; 2020); Huang et al. (2023) from OpenCompass OpenCompass (2023), a large language model evaluation benchmark. We collect the evaluation results of 32 widely used LLMs (e.g. LLAMA (Touvron et al., 2023a), Qwen (Bai et al., 2023), ChatGLM (Zeng et al., 2023), Gemma (Mesnard et al., 2024)). We select the first 21 LLMs based on their release dates as the initial LLMs and use the remaining 11 LLMs as the new LLMs to be tested.

We do the toy experiment with ratio 0.1, which is described in section 5.

### C.2   HYPOTHESES TESTING

To verify the hypothesis that there is a gap between the semantic embedding space and the evaluation results space, we plot Figure 7 on one task of the MMLU benchmark. The details of plotting Figure 6 are as follows:

First, we select the k-nearest neighbors in the semantic space for each sample in a specific sub-scenario and compute the distances between them.

Next, we calculate the distances between the corresponding sample pairs in the performance space.

Finally, we use these two distances as the x and y coordinates to plot a scatter plot.

Figure 7 is a scatter plot for a randomly selected sub-scenario. We can observe that the scatter plots for almost all sub-scenarios form a dense cluster without any clear relationship, which proves that it is not effective to directly use the semantic information of pre-trained models for efficient evaluation.

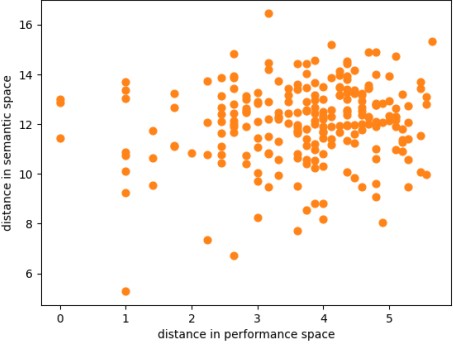

Figure 7: **Distances in Two Spaces for Nearest Neighbor in Semantic Space.** Each point represents a nearest-neighbor pair in the semantic space. The vertical coordinate indicates the distance between the nearest neighbors in the semantic embedding space, while the horizontal coordinate represents the distance between the corresponding samples in the evaluation results space.

# D SOMETHING FOR BINARY EVALUATION MATRIX

If the ME in subsection 3.1 yields binary results of 0 or 1, in other words, the evaluation results matrix $D_i$ in subsection 3.2 is a binary matrix, then some equations can be calculated using alternative methods or may require further steps.

## D.1 ANOTHER WAY TO CALCULATE IMPORTANT SCORE

With the ME yields binary results of 0 or 1, we can alternatively calculate $v(x|F)$ in subsection 4.1 using the quantity difference between 0 and 1 values shown in Equation 11.

$$v(x|F) = \frac{1}{|\sum_{m=1}^{M}[\mathbb{I}(\hat{y}^{(f_m)} = y) - \mathbb{I}(\hat{y}^{(f_m)} \neq y)]|}. \tag{11}$$

where $\mathbb{I}(\cdot)$ denotes the indicator function, which takes the value 1 if the condition inside the parentheses is true, and 0 otherwise.

## D.2 FURTHER STEPS FOR OPTIMAL TRANSPORT

If $D_i$ in subsection 4.2 is a binary matrix, further steps should be done after getting matrix $D_i^{(sy)}$ in subsection 4.2. Specifically, the elements in $D_i^{(sy)}$ should be divided into 0-1 by 0.5 as the threshold.

# E THE PSEUDO-CODE OF OUR METHOD

---

**Stage 1** An iterative sampling process for a specific task

---

**Inputs:** Task $T_i$, initial LLMs $F_B$, target LLM $f_t$, evaluation results $D_i$ of $F_B$, desired sample size $|T_i^s|$, number of similar models $n$, and instance increment per iteration $d$
**Output:** A selected subset $T_i^s \subseteq T_i$
$P \leftarrow$ initialize probe set using Equation 6
**while** $|P| < |T_i^s|$ **do**
    $d_t^P \leftarrow$ evaluate $f_t$ on $P$
    $D_{i,P} \leftarrow$ extract from $D_i$ the results corresponding to $P$
    $F_S \leftarrow$ retrieve top-$n$ models most similar to $f_t$ based on $D_{i,P}$
    $P_d \leftarrow$ select $d$ new instances not in $P$ using Equation 7
    $P \leftarrow P \cup P_d$
**end while**
$T_i^s \leftarrow P$
**return** $T_i^s$

---

**Stage 2** Performance prediction process for a specific task

---

**Inputs:** Task $T_i$, initial LLMs $F_B$, target LLM $f_t$, all evaluation results $\{D_1, ..., D_N\}$ of $F_B$, a subset benchmark $\{T_1^s, ..., T_N^s\}$
**Output:** Estimated performance $\hat{p}_i$ for task $T_i$
$\{\mathbf{v}_1, ..., \mathbf{v}_N\} \leftarrow$ task $T_i$'s mean performance vector
$T_i^{st}, D_i^{st} \leftarrow T_i$'s most similar tasks and corresponding selected instances
$T_i^o, D_i^o \leftarrow$ task $T_i$'s oversampled instances using Equation 9
$c_{i0}^{ns} \leftarrow$ get $f_t$'s estimated performance on low importance instances using Equation 10
$c_{i1}^{ns}, c_{i2}^{ns} \leftarrow$ get $f_t$'s estimated performance on remaining instances using collaborative filtering
$\hat{c}_i^{ns} \leftarrow \hat{c}_{i0}^{ns} + \hat{c}_{i1}^{ns} + \hat{c}_{i2}^{ns}$
$\hat{p}_i \leftarrow$ compute by replacing $a_i^{ns}$ in Equation 8 as $\hat{c}_i^{ns}$
**return** $\hat{p}_i$

---

## F    INSTANCE SAMPLING DETAILS

Assume $|T_i|$ is the number of test instance in the dataset for the $i$-th task, and let $\alpha$ be the sampling ratio, then $\alpha \times |T_i|$ represents the number of sampled instances. To ensure an accurate estimation of LLM performance, we set a minimum sample size of 20 for each dataset. When $|T_i|$ is less than 20, we will use all samples from the current dataset. When $|T_i|$ is greater than or equal to 20 but $\alpha \times |T_i|$ is less than 20, we will set the number of sampled instances for the current dataset to 20.

Table 4: **Methods at Different Ratios.**

| Ratio | Method | tok-k | NDCG |
|-------|--------|-------|------|
| **0.1** | Our | 0.5839 | 0.9983 |
| | Tinybenchmark | 0.5942 | 0.9979 |
| | Sort&Search | 0.4548 | 0.9959 |
| **0.2** | Our | 0.6161 | 0.9984 |
| | Tinybenchmark | 0.6135 | 0.9981 |
| | Sort&Search | 0.5323 | 0.9967 |
| **0.3** | Our | 0.6903 | 0.9987 |
| | Tinybenchmark | 0.6471 | 0.9983 |
| | Sort&Search | 0.6532 | 0.9985 |
| **0.4** | Our | 0.7306 | 0.9989 |
| | Tinybenchmark | 0.6845 | 0.9986 |
| | Sort&Search | 0.6952 | 0.9987 |
| **0.5** | Our | 0.7484 | 0.9992 |
| | Tinybenchmark | 0.7345 | 0.9989 |
| | Sort&Search | 0.7161 | 0.9991 |

## G    OTHER METRICS

We calculated the Top-k Recall and NDCG metrics based on the accuracy of both historical models and new models on each sub-scenario and then averaged the results. For historical models, we directly used their actual accuracies, while for new models, we used the estimated accuracies.

We present the detailed results on the LB benchmark, as shown in Table 4, which demonstrate that our method still holds a significant advantage.

## H    ABLATION STUDY

### H.1    OPTIMAL TRANSPORT MODULE

We plot the MAE metric on different ratios on MMLU. Figure 8 shows the result that our method with optimal transport is consistently better than that without optimal transport. All in all, introducing information from other tasks into efficient evaluation methods can improve method performance.

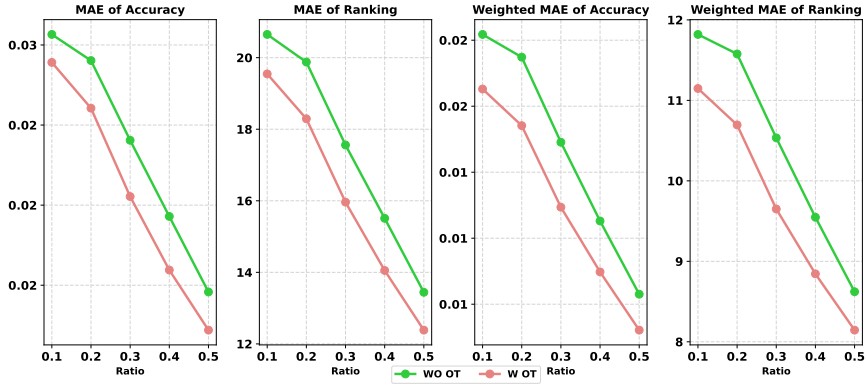

Figure 8: **The MAE and Weighted MAE of Method with OT and Method without OT.**

### H.2    TRADE-OFF BETWEEN PERFORMANCE AND COST

The collaborative filtering method used in this paper is the simplest similarity-based approach. As introduced in section 2, there are many other collaborative filtering methods, such as matrix factorization, neural collaborative filtering, and graph neural network-based collaborative filtering. While these methods can certainly be incorporated into our framework, they typically require significantly more computation time compared to the similarity-based approach. We do matrix factorization which is the same as the method called UCBE-LRF Zhou et al. (2025) on lb benchmark with 0.1 ratio. The result is shown in Table 5.

Table 5: **Comparisons under Ratio 0.1.**

| Method | Ratio | MAE Acc | Weighted MAE Acc | MAE Rank | Weighted MAE Rank | Total Time (s) |
|--------|-------|---------|------------------|----------|-------------------|----------------|
| UCBE-LRF | 0.1 | 0.0253 | 0.0177 | 20.68 | 13.73 | ≥5400 |
| OUR | 0.1 | 0.0288 | 0.0177 | 25.79 | 15.72 | 25.2 |

We can observe that while the MAE Acc improves by only 0.003, the total time increases by at least dozens of times, which is unacceptable.

### H.3   HYPERPARAMETER

Next, we will analyze the hyperparameters involved in the method in section 4. They are $|S|$, $\alpha$, number of iterations in subsection 4.1 and $\tau_0$, $\tau_1$, $\tau_2$, $q$ in subsection 4.2. Here $\tau_2$ and $q$ are hyperparameters related to $\tau_3$ in subsection 4.2. Specifically, $\tau_3 = \max(\tau_2, [\overline{\mathbf{c}}_i]_q)$. Here $[\overline{\mathbf{c}}_i]_q$ refers to the $q$ quantile of $\overline{\mathbf{c}}_i$ and $\overline{\mathbf{c}}_i$ a vector formed by the average similarities of different tasks.

The specific roles of these hyperparameters are shown in Table 6.

Figure 9 shows the MAE and

Table 6: **Roles of Different Hyperparameters.**

| Hyperparameter | Roles |
|----------------|-------|
| $|S|$ | The number of similar models for a new target model. |
| $\alpha$ | A hyperparameter that determines the importance score of the sample. |
| Number of iterations | The number of iterations in the instance selection process. |
| $\tau_0$ | A hyperparameter used to select similar tasks. |
| $\tau_1$ | A hyperparameter used to determine unimportant instances. |
| $\tau_2$ | A hyperparameter used to determine the prediction mode. |
| $q$ | Another hyperparameter used in prediction mode decision. |

weighted MAE with different hyperparameters on the MMLU benchmark. Each row of the figure represents the performance change if only one hyperparameter is changed and the rest are unchanged. From top to bottom, they represent the $|S|$, number of iterations, $\alpha$, $q$, $\tau_2$, $\tau_1$ and $\tau_0$, respectively. From the figure, we can find that our method is robust to different hyperparameters.

## I   THE USE OF LARGE LANGUAGE MODELS

We used a large language model (LLM) solely as a writing assistant to improve the clarity and readability of the manuscript (e.g., polishing grammar and phrasing). The LLM was not involved in research ideation, experimental design, implementation, or analysis. All scientific contributions and results are entirely the work of the authors.

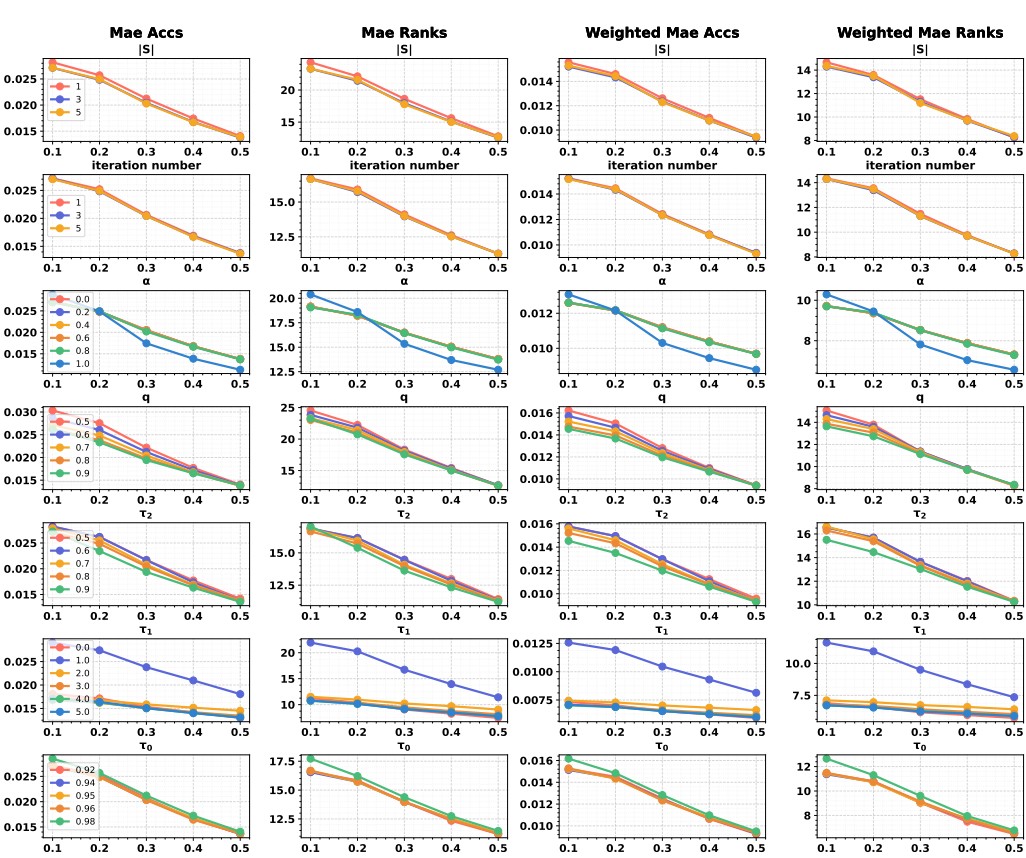

Figure 9: **The MAE and Weighted MAE with Different Hyperparameters.** Each row represents a hyperparameter, and each column corresponds to a metric of efficient evaluation methods. Our method is robust to hyperparameter variations.

