# OpenReview forum: "Efficient Evaluation of Large Language Models via Collaborative Filtering"
_ICLR.cc/2026/Conference — Submitted to ICLR 2026_

### Official Review · Reviewer_ddJ9 · 2025-10-29

**Soundness:** 2
**Presentation:** 4
**Contribution:** 2
**Rating:** 2
**Confidence:** 4

**Summary:**

The paper introduces a method to efficiently evaluate large language models (LLMs) using collaborative filtering. The proposed approach consists of two stages: instance sampling and performance prediction. Experiments conducted on multiple LLMs and benchmarks demonstrate that the method performs well.

**Strengths:**

1. The paper is well-motivated. How to evaluate llm efficiently is very valuable.
2. The Experiments on multiple LLMs and benchmarks all shows the method works well.
3. This paper is readable.

**Weaknesses:**

1. The method still relies on **B early LLMs** that have already been evaluated on the entire benchmark. However, in real-world scenarios, it may be difficult to directly obtain the outputs of these B models on the benchmark. Therefore, we may first need to **generate** the responses of these B models on the whole benchmark. From this perspective, should we also **take the cost of these B models into account**?
2. New models are continuously emerging and being updated, but should the **B early LLMs** also be updated accordingly? If not, the selected samples may lose relevance for newer models; however, if they do need to be updated, each updated model would have to generate responses for the entire benchmark again, which would undermine the efficiency of the proposed method.

**Questions:**

1. Could you describe in detail the experimental setup shown in Figure 2?
2. Section 3.3 presents a simple yet effective evaluation method, but how is it related to the approach proposed in Section 4?
3. Figure 3 shows that the *Random Selection 0.1* setting took the least amount of time (0.0023 s). Based on this estimation, would testing all the data take less than 0.023 s? Is this calculation correct? Could you provide the actual time required to evaluate all the data?
4. One of the key aspects of an evaluation method is having a unified standard. However, this method does not specify certain parameters, such as the sample ratio or **B** (the number of early models), which may lead to inconsistencies or confusion when comparing results.

---

> ### Author Response · Authors · 2025-11-13
>
> ---
> **Q1**:Concerns regarding the acquisition of early models.
>
> ---
> **Answer**:
>
> We appreciate the reviewer’s thoughtful comment.
> First, we would like to clarify that almost all existing efficient evaluation methods (except for purely random sampling baselines) rely on the performance records of a set of historical models on public benchmarks. Similarly, in our case, the results of these early models are directly obtained from openly available benchmark leaderboards such as OpenCompass, HuggingFace Open LLM Leaderboard, and MMLU, which already contain full evaluation results for hundreds of models. Therefore, the cost of obtaining these early-model results is negligible.
>
> We acknowledge that the early-model library may need to be updated over time. However, our experimental setup has already taken this factor into account: we explicitly sorted models by their release date (with time spans up to 6 months) and used earlier models as the initial set. **This design empirically examines whether older models remain informative for evaluating newer ones**, and the results (Section 5) confirm the robustness of our method under this setting.
>
> Finally, we emphasize that our method targets resource- or time-constrained evaluation scenarios. When sufficient resources become available, one can incrementally refresh the early-model library, maintaining both practicality and efficiency.
>
> ---
> **Q2**: Details settings of Figure 2.
>
> ---
>
> **Answer**:
>
> Thank you for the question. We would like to further clarify that **Figure 2 illustrates an example from one specific subtask in the MMLU experiment described in Section 5**, which is intended to intuitively demonstrate how our method predicts performance at the single-task level. We have made this explanation clearer in the revised version of the paper to ensure readers can accurately interpret the figure. It is worth noting that the trend shown in this example is consistent across different tasks, confirming the generality and stability of our proposed method.
>
> ---
> **Q3**: The relationship between Section 3.3 and Section 4.
>
> ---
>
> **Answer**:
>
> Thank you for the question. The relationship between Section 3.3 and Section 4 can be summarized as follows:
>
> As discussed in the first and last paragraphs of Section 3.3, the simple baseline was designed to **demonstrate that semantic information of benchmark instances cannot be directly leveraged for efficient evaluation**. Even without using semantic embeddings, high prediction accuracy can be achieved purely from historical evaluation results. This observation motivates our decision in Section 4 to discard semantic features entirely.
>
> The experimental results in Section 3.3 (as shown in Figure 5) also highlight the limitation of previous methods: none of them consistently outperform this simple baseline. In contrast, our two-stage collaborative filtering method proposed in Section 4 consistently and stably surpasses the baseline across all benchmarks, **demonstrating its robustness and superiority in both accuracy and consistency.**
>
> ---
>
> **Q4**: Regarding the concern about time calculation.
>
> ---
>
> **Answer**:
>
> Thank you for the reviewer’s question. We would like to clarify that the time reported in Figure 3 does not include the model inference time (as also noted in Section 5), but only covers the time for sample selection and performance extrapolation. **Our purpose was to compare the algorithmic efficiency of different methods.** **Since the average token length of the selected samples is roughly the same across methods at the same sampling ratio (we have indeed conducted the statistics), we can reasonably assume that the average inference time per instance remains constant.** Therefore, the reported numbers reflect the computational overhead of the methods themselves, rather than the total end-to-end evaluation time.
>
> ---
>
> **Q5**: Concern about certain parameters.
>
> ---
>
> **Answer**:
>
> Thank you for the question. We would like to clarify that **all parameter settings are explicitly presented in Section 5**. Specifically, the sample ratio is described in the Setups subsection, and the number of early models (B) is detailed in the Benchmark subsection. We can **ensure that all method comparisons are fair and reliable**. In addition, the x-axis of Figure 5 directly corresponds to the sample ratio, and our method consistently performs well across all ratio settings, demonstrating its robustness and stability.

---

> > ### Comment · Reviewer_ddJ9 · 2025-11-20
> >
> > Thanks for your reply. Most of my concerns have been addressed. But could you list and compare the other methods mentioned in the Q1 response?

---

> > > ### Author Response · Authors · 2025-11-20
> > >
> > > Dear Reviewer,
> > >
> > > Thank you for the reviewer’s response. Below, we provide the methods mentioned in Question 1 to illustrate why these methods require historical samples:
> > >
> > > ---
> > > 1. Anchor Points Method: This method uses the confidence scores of historical models to cluster the samples and obtain subsets of the data. The target model then performs inference on these subsets and extrapolates its overall performance. However, confidence scores are difficult or impossible to obtain for certain closed-source models, so we did not include this method in our comparison. Nevertheless, the core idea of this baseline is consistent with that of the paper.
> > >
> > > ---
> > > 2. TinyBenchmark: This method requires access to historical models’ predictions on the full dataset to train an auxiliary evaluation model, which is used to select the representative subset and perform performance extrapolation. When a new target model arrives, it only needs to be evaluated on this subset, and its overall performance is then inferred by combining the subset predictions with the learned evaluation model. The limitations of this design and corresponding empirical comparisons are discussed in Sections 4.3 and 5.
> > >
> > > ---
> > > 3. Sort\&Search: This method also requires historical models’ predictions on the samples to estimate sample difficulty, but its sampling and extrapolation procedures are relatively simpler; we likewise provide analysis and comparisons in Sections 4.3 and 5.
> > >
> > > ---
> > > 4. UCB-E / UCB-E-LRF: The two methods proposed in this paper are primarily designed for model selection and can be applied to our efficient evaluation setting. The first method does not require historical data, but in essence behaves similarly to random sampling. The second method leverages matrix factorization, which fundamentally requires the results of historical models.
> > >
> > > ---
> > > The above methods rely on historical model results to varying degrees, and we believe this is reasonable and something researchers should make full use of. In an era where large language models evolve rapidly, a substantial amount of public evaluation data on past models is continuously accumulating. If we merely view these results statically, they will gradually lose value as new models emerge; however, in our setting, these seemingly “outdated” results can instead be repurposed to provide valuable assistance for efficient evaluation.
> > >
> > > We sincerely hope that the above explanation alleviates your concerns, and we look forward to receiving your recognition. If you find our clarifications and responses satisfactory, we would be deeply grateful if you could reflect this in your evaluation and support the acceptance of our paper.
> > >
> > > Thank you once again for your valuable feedback and for taking the time to review our work.
> > >
> > > ---
> > > Reference Papers:
> > >
> > > [1] Anchor points: Benchmarking models with much fewer examples. In EACL, 2024.
> > >
> > > [2] tiny-benchmarks: evaluating llms with fewer examples. In ICML, 2024.
> > >
> > > [3] Life-long benchmarks: Efficient model evaluation in an era of rapid progress. In ICLR, 2024.
> > >
> > > [4] On speeding up language model evaluation. In ICLR, 2025.

---

> > > > ### Comment · Reviewer_ddJ9 · 2025-11-20
> > > >
> > > > Thanks for your reply. I increase the score to 4.

---

### Official Review · Reviewer_aKUQ · 2025-10-30

**Soundness:** 3
**Presentation:** 3
**Contribution:** 3
**Rating:** 6
**Confidence:** 3

**Summary:**

This paper proposes an efficient LLM evaluation method by reframing the task as a collaborative filtering problem. Specifically, each model is treated as a user, and each test example as an item. The model performance is regarded as a rating. By selecting a small amount of high-variance subset of samples (i.e., Probe Set) and using optimal-transport–based alignment to predict results on unseen samples, the method achieves nearly full-set accuracy with much lower evaluation cost.

**Strengths:**

**Novel application of collaborative filtering for model evaluation.**
The paper introduces a new perspective by framing LLM evaluation as a collaborative filtering problem. Instead of introducing a new metric or dataset, it contributes a new perspective for efficient model evaluation. This conceptual shift opens a promising direction for accelerating LLM evaluation efficiently.

**Simplicity and practicality.**
The approach is straightforward to implement. It only requires existing evaluation records for instance selection. No additional model training or annotation is needed, making it easily applicable in real-world evaluation pipelines.

**Clarity and readability.**
The paper is well-written and easy to understand. The motivation, methodology, and experimental setup are easy to follow. I appreciate the informative illustrations, such as Figure 2, Figure 3, and Figure 4, making the background and method clearer to me.

**Weaknesses:**

The method assumes that new models share similar capability structures with historical ones. Performance may degrade on out-of-distribution models. For example, there are several expert models, which may excel in one specific domain such as math. In this case, the performance of this model is not uniformly distributed like general models. It is not clear whether the proposed method also holds for such domain-specific models.

**Questions:**

- What is abab-6.5? It seems that this model only occurs once in the introduction without any description of it.

- I am curious about the implementation details of the "historical" baseline in Table 1. It seems that details about it are missing in Section 3.3.

- What is the model sim distribution in step 2 of stage 1?

---

> ### Author Response · Authors · 2025-11-13
>
> ---
> **Q1**: Conern about domain specificed model.
>
> ---
>
> **Answer**:
>
> We acknowledge that this is indeed a valid concern. Therefore, we suggest that if one aims to obtain more accurate performance estimation, it is preferable to construct the historical model library using domain-specific models corresponding to the target domain. However, **when the goal is model ranking, our method remains effective**, since stronger domain models naturally perform better on discriminative instances.
>
> Moreover, **we must also note that the performance gap between general-purpose models and domain-specific models has been rapidly narrowing**. For instance, in the mathematics domain, most top-performing models on current leaderboards (such as Grok-4, Qwen3-235B-A22B-Thinking and GPT-5 and so on) can no longer be clearly classified as domain-specific models, reflecting the convergence of general capabilities across modern LLMs. We believe this also, to some extent, demonstrates the reasonableness of including only general-purpose models as part of the historical model library.
>
> ---
> **Q2**: Concern about abab-6.5.
>
> ---
> **Answer**:
>
> Thank you for the reviewer’s question. The **abab-6.5** model refers to a model developed by **MiniMax**, belonging to the **abab-6.5 model family**. In our paper, it was mentioned only as an example to illustrate the performance differences across models, which is why we did not provide further details in later sections. **We have added a clarification in the revised paper to indicate the companies behind the models, making it easier for readers to understand their origins.**
>
> ---
> **Q3**: Concern about "historical baseline".
>
> ---
> **Answer**:
>
> Thank you for the question. The “historical baseline” refers to the baseline method described in Section 3.3. **The only difference from** the semantic embedding–based baseline is that, instead of using semantic embeddings, **it constructs embeddings by concatenating the historical models’ results on each sample.** This is explained at the beginning of the third paragraph in Section 3.3. We acknowledge that the current description may cause confusion, and **we have revised the paper to make this point clearer.**

---

### Official Review · Reviewer_TJ7U · 2025-10-31

**Soundness:** 3
**Presentation:** 2
**Contribution:** 3
**Rating:** 6
**Confidence:** 3

**Summary:**

This research studies the efficient evaluation problem for large language models. In particular, the researchers consider the evaluations of new LLMs by using the evaluation results from older ones. The authors propose to study this problem by considering the collaborative filtering framework, where LLMs are users and samples are items. The empirical results on the real-world datasets demonstrate that the proposed method can more precisely predict model performance than other baselines.

**Strengths:**

1. The research question of evaluating LLMs' performance with a limited inference budget is interesting.
2. The proposed idea of considering it as a collaborative filtering problem is reasonable.
3. The ablation study on running time makes the experiment more rigorous.

**Weaknesses:**

I'm happy to see that you consider the model-sample matrix as the user-item matrix. I also agree with providing some seed samples to testify that the model can solve the "cold-start" problem. But I have these two questions:
1. After collecting the performance on those "cold-start" samples, why can we not simply apply matrix factorization methods (the simplest approach for recommendation under the collaborative filtering setting) to directly reconstruct the missing scores from this new model in the matrix?
2. The old LLMs have provided full predictions on all test samples, which is different from the recommendation settings, where even activated users cannot provide feedback to all items. So, I don't think the current method leverages this unique property.

In the experimental setting, why do we split the candidate models by release date? What impact will happen if we don't have that information?

Minors:
1. Line 185: The cite format is improper.

**Questions:**

Please see the weaknesses.

---

> ### Author Response · Authors · 2025-11-13
>
> ---
> **Q1**: Concern about matrix factorization.
>
> ---
> **Answer**:
>
> Thank you for the question. The reason we do not directly apply matrix factorization is due to its computational cost. Performing matrix factorization for each task would be extremely expensive. **As shown in Appendix H.2, we included a comparison method named UCBE-LRF, which adopts matrix factorization.** While it indeed achieves slightly better accuracy, its **time cost increases dramatically**, making it unsuitable for the goal of efficient evaluation. **Considering both evaluation efficiency and accuracy, we believe our proposed method provides a more practical and balanced solution.**
>
> ---
> **Q2**: Concern about not fully leveraging the unique property that old LLMs provide complete predictions on all test samples.
>
> ---
> **Answer**:
>
> Thank you for the reviewer’s valuable suggestion. We agree that this unique property could potentially be further exploited in other ways. Here, we would like to clarify **how our current approach already leverages this property**:
>
> 1. In traditional recommendation systems, the number of users must be very large because the user–item interaction matrix is highly sparse. In contrast, **our setting benefits from the full prediction coverage of old LLMs**, **allowing us to achieve reliable estimation using only dozens or hundreds of models**—this directly reflects our utilization of the *complete evaluation* property (as discussed in Appendix B).
> 2. Beyond this, our method also leverages another important feature—**the controllable sample selection process**. In recommendation systems, a user’s historical interactions are typically random and uncontrollable, whereas in our setting, we can deliberately obtain model responses on selected samples, which provides stronger supervision for evaluation.
>
> We appreciate the reviewer’s insightful comment and welcome further discussion. We also acknowledge that, while our current approach already considers these characteristics, there may be additional properties or more effective ways to exploit them in future work.
>
> ---
> **Q3**: Concern about spliting the candidate models by release date
>
> ---
>
> **Answer**:
>
> We thank the reviewer for the question. We adopt the time-based split because **it reflects a realistic scenario in which new models are introduced progressively, leading to a natural shift in the distribution of model capabilities over time.** Such temporal drift makes the time-based split more challenging than a random split. The fact that our method maintains strong performance under this shifted setting further demonstrates that **it remains effective even without relying on release-date information.**
>
> ---
> **Q4**: Line 185: The cite format is improper.
>
> ---
> **Answer**:
>
> Thank you for the reviewer’s suggestion regarding citation formatting. We have modified the paper and will carefully review all other references to ensure consistency and accuracy throughout the paper.

---

> > ### Comment · Reviewer_TJ7U · 2025-11-19
> > **Official Comments to Author's Response**
> >
> > I am happy to see these discussions, and my concerns have been addressed. Please include the discussions of Q1 and Q2 in the manuscript.

---

> > > ### Author Response · Authors · 2025-11-20
> > >
> > > Dear Reviewer,
> > >
> > > We sincerely appreciate your positive acknowledgment. We have incorporated your suggestions into the manuscript, and added the necessary guidance in the main text to direct readers to the corresponding appendices. If we understand correctly, this indicates that our rebuttal and subsequent responses have adequately addressed your concerns. We would be truly grateful if you could reflect this in your final evaluation and consider supporting the acceptance of our paper.
> > >
> > > Thank you once again for your valuable feedback and for taking the time to review our work.

---

### Official Review · Reviewer_fDsg · 2025-11-01

**Soundness:** 1
**Presentation:** 2
**Contribution:** 2
**Rating:** 2
**Confidence:** 3

**Summary:**

This paper studies the efficient evaluation of LLMs, an important and practical problem in model assessment. The authors identify a limitation of existing efficient evaluation methods, namely their inability to generalize to task-specific scenarios. To address this, they propose a two-stage evaluation framework inspired by collaborative filtering, offering a new perspective on sample selection for efficient evaluation. Empirical results demonstrate the effectiveness of the proposed approach.

**Strengths:**

1. The paper tackles a timely and practical problem in LLM evaluation.

2. The use of collaborative filtering to address sample selection and performance prediction is novel and provides an interesting perspective.

3. Extensive experiments are conducted to validate the effectiveness of the proposed method.

**Weaknesses:**

1. The motivation is insufficiently developed. The paper does not clearly explain how existing efficient evaluation methods fail to capture task-specific performance and model ranking.

2. The discussion of related work is limited and lacks depth. At least clear descriptions of how prior works select samples should be discussed and compared in the paper.

3. Certain design choices in the proposed method appear questionable (see Questions for details).

**Questions:**

1. In the proposed data selection process, both easy and difficult samples are excluded. While removing easy samples is reasonable, excluding difficult samples seems questionable, as they often reveal the performance gap among models. If initial models are used to estimate difficulty, the estimation may become inaccurate as the models evolve. Could the authors discuss this potential issue?

2. The baseline design relies on two assumptions (lines 196–202) that may not always hold. The first assumes that models produce consistent responses for semantically similar samples. However, semantically similar questions can differ significantly in difficulty. The second assumes that models behave consistently within the same cluster. Empirical evidence supporting these assumptions would strengthen the paper.

3. The idea of selecting similar models is reasonable, but the instance selection based on “variance” in model responses is unclear. It is not evident how this aligns with the collaborative filtering principle that similar users have similar interactions. Could the authors elaborate on the rationale for choosing samples with high variance among similar models?

---

> ### Author Response · Authors · 2025-11-13
>
> ---
> **Q1**: The paper lacks explanations of why existing methods fail to capture task-specific performance and model rankings.
>
> ---
> **Answer**:
>
> We thank the reviewer. We have **incorporated our responses to Q1 into the revised paper**. We would like to clarify that previous efficient evaluation methods primarily focused on **overall benchmark performance**, without addressing **task-level performance differences**. Our paper is, to our knowledge, **the first to explicitly emphasize the importance of task-specific evaluation**. As discussed in the introduction, this focus is well-motivated—for example, two models with the same average code score may differ significantly, with one performing better in Python and the other in Java. Moreover, **task-level evaluation is inherently more challenging than overall evaluation**: Prior methods emphasize overall performance but neglect task-level accuracy, leading to suboptimal and biased results, especially on tasks with fewer samples.
>
> ---
> **Q2**: Need clear descriptions of how prior works select samples.
>
> ---
> **Answer**:
> To clarify, the prior works differ in how they select samples:
> Lifelong Benchmark performs uniform sampling across instance difficulty level.
> TinyBenchmark selects samples around cluster centers.
> UCBE adopts a random uniform sampling strategy guided by a multi-armed bandit process.
>
>
> **Moreover, we emphasize that sample selection and performance estimation are tightly coupled stages—they cannot be analyzed independently nor arbitrarily combined across methods.** For instance, Lifelong and UCBE employ simple extrapolation, while TinyBenchmark relies on a neural network that is comparatively computationally heavy.
>
> Considering both simplicity and effectiveness, **our proposed approach achieves superior balance** between efficiency and accuracy in performance estimation. We **have modified the appendix in paper to show more details of previous works**
>
> ---
> **Q3**: Concern about discarding samples that are too difficult.
>
> ---
> **Answer**:
>
> We would like to clarify one of our underlying assumptions: model capabilities evolve progressively over time. **When the temporal gap between earlier and newer models is not large—as in our experimental setup**, where models are sorted by release date with a maximum span of about 6 months—our results already demonstrate that the proposed method remains robust under such temporal variation.
>
> That said, we agree that as models continue to advance, the notion of “difficult” samples may gradually shift. Therefore, we **recommend incrementally updating or expanding the historical model set to reflect the evolving capability landscape.** As newer models are incorporated, the definition of difficult instances will naturally adapt to align with current model performance levels. **These updates can be efficiently obtained from publicly available evaluation results collected through open benchmark repositories or automated web crawlers.**
>
> ---
> **Q4**: Concern about assumptions of baseline.
>
> ---
>
> **Answer**:
>
> We would like to clarify that **our proposed baseline does not rely on the two assumptions mentioned by the reviewer.** In fact, **the main purpose of Section 3.3 is precisely to demonstrate that the first assumption—that models produce consistent responses for semantically similar samples—does not hold in practice**, thereby motivating our decision **not to use semantic embeddings in the baseline method**.
>
> Regarding the second assumption, we note that our baseline **performs clustering on the embeddings constructed from historical model performance**, rather than on semantic features. Since these embeddings inherently encode models’ behavioral patterns over benchmark instances, **the clustering process naturally satisfies the second assumption to a reasonable extent.**
>
> We acknowledge that this section may cause confusion. You can also see **our answers for Q3 of Reviewer aKUQ. We have modified the papaer.**
>
> ---
> **Q5**: Concern about choosing samples with high variance among similar models.
>
> ---
>
> **Answer**:
>
> In collaborative filtering, the basic assumption is that similar users tend to exhibit similar behaviors—for example, liking or disliking similar items. In this context, when we identify several similar users, an item on which these users show disagreement or divergence becomes particularly informative: **it serves as a discriminative signal that helps refine the user profiles.** To better understand a target user, the most straightforward approach is to test their preference on precisely such an item.
>
> Analogously, in our setting, instances with **high variance among similar models’ responses play the same role.** They effectively **help us differentiate the target model from its most similar historical models**, enriching the characterization of the model’s capability space and leading to **more accurate ranking and evaluation outcomes.**

---

### Meta-Review · Area_Chair_84Jz · 2026-01-06

**Summary:**

While reviewers acknowledged the novelty of applying Collaborative Filtering to LLM evaluation and praised the paper's clarity, the final decision to reject was driven by critical methodological flaws that remained unresolved despite the authors' rebuttal. The core assumption that new models share capability structures with historical ones was deemed too idealistic, posing significant risks for expert or out-of-distribution models. A major point of contention was the strategy of excluding "hard samples," which reviewers argued would obscure true performance gaps. Furthermore, the proposed framework treats historical data as sparse despite the availability of full predictions, and the authors' defense of "simplicity" did not technically justify ignoring this full information or preferring their approach over standard Matrix Factorization. Although the authors successfully clarified concerns regarding data acquisition costs, the fundamental deficiencies regarding the validity of the sampling strategy, the underutilization of historical data, and the weak motivation for why existing methods fail persisted.

**Reviewer Concerns:**

Cost of Historical Data: The authors clarified that historical model data can be sourced from public leaderboards, alleviating concerns regarding data acquisition costs and practicality.

Limitations of the Core Assumption: Although the authors claimed validity within a 6-month window, this does not theoretically resolve the issue that expert models or models with significant architectural shifts may not align with historical distributions.

Choice of Algorithm: The authors explained that the simpler method was chosen to balance computational efficiency with performance and to serve as a foundational baseline for this new intersection of fields.

**Reviewer Scores:**

none

---

### Decision · Program_Chairs · 2026-01-26

Reject